# Designing Electrical Stimulation Platforms for Neural Cell Cultivation Using Poly(aniline): Camphorsulfonic Acid

**DOI:** 10.3390/polym15122674

**Published:** 2023-06-14

**Authors:** Fábio F. F. Garrudo, Robert J. Linhardt, Frederico Castelo Ferreira, Jorge Morgado

**Affiliations:** 1Instituto de Telecomunicações, Instituto Superior Técnico, Universidade de Lisboa, Avenida Rovisco Pais, 1049-001 Lisbon, Portugal; 2Department of Bioengineering, Instituto Superior Técnico, Universidade de Lisboa, Avenida Rovisco Pais, 1049-001 Lisbon, Portugal; frederico.ferreira@tecnico.ulisboa.pt; 3iBB—Institute for Bioengineering and Biosciences, Instituto Superior Técnico, Universidade de Lisboa, Avenida Rovisco Pais, 1049-001 Lisbon, Portugal; 4Associate Laboratory i4HB, Institute for Health and Bioeconomy, Avenida Rovisco Pais, 1049-001 Lisbon, Portugal; 5Department of Chemical and Biological Engineering, Biology and Chemistry and Chemical Biology, Center for Biotechnology and Interdisciplinary Studies, Rensselaer Polytechnic Institute, 110 8th Street, Troy, NY 12180, USA; linhar@rpi.edu

**Keywords:** bioelectricity, biophysical cues, additive manufacturing, electroconductive polymers, neurological diseases, nerve/neuron regeneration, voltage-dependent calcium channels

## Abstract

Electrical stimulation is a powerful strategy to improve the differentiation of neural stem cells into neurons. Such an approach can be implemented, in association with biomaterials and nanotechnology, for the development of new therapies for neurological diseases, including direct cell transplantation and the development of platforms for drug screening and disease progression evaluation. Poly(aniline):camphorsulfonic acid (PANI:CSA) is one of the most well-studied electroconductive polymers, capable of directing an externally applied electrical field to neural cells in culture. There are several examples in the literature on the development of PANI:CSA-based scaffolds and platforms for electrical stimulation, but no review has examined the fundamentals and physico-chemical determinants of PANI:CSA for the design of platforms for electrical stimulation. This review evaluates the current literature regarding the application of electrical stimulation to neural cells, specifically reviewing: (1) the fundamentals of bioelectricity and electrical stimulation; (2) the use of PANI:CSA-based systems for electrical stimulation of cell cultures; and (3) the development of scaffolds and setups to support the electrical stimulation of cells. Throughout this work, we critically evaluate the revised literature and provide a steppingstone for the clinical application of the electrical stimulation of cells using electroconductive PANI:CSA platforms/scaffolds.

## 1. Introduction

Neurological diseases, in particular neurodegenerative ones, are chronic, cause incapacity and their progression cannot be stopped or reversed. The onset of these diseases takes place in the brain tissue, in which many factors associated with cell senescence/death homeostasis are thrown out of balance. In these diseases, the basic neurological functions are compromised by massive or localized progressive cell death. Still, most symptoms only start to appear later. Due to the incapacity of the brain tissue to regenerate, patients afflicted with these diseases have no hope for a cure. Palliative treatments dealing with behavioral symptoms, motricity and pain are available. However, as these diseases progress, medication loses its effectiveness.

Electrical stimulation has emerged as a powerful therapeutic tool for the treatment of neurological diseases. Through deep-brain stimulation (DBS), it is possible to apply direct current into the brain tissue. This technique has numerous beneficial effects on the brain tissue, including on: (1) neurogenesis; (2) neurite ramification; (3) axonal remodeling; (4) synaptogenesis; (5) gliogenesis; (6) increase in neuronal size; and (7) increase in vascularization.

Some trials have shown evidence of positive effects on patients after treatment. In Alzheimer’s disease patients, Sankar and colleagues [1] succeeded in retarding brain shrinkage. This was attributed to a reduction in Amyloid beta (Aβ) deposition and improved memory, as Mann and colleagues [2] observed for mice when DBS was applied for 25 days. Other trials were performed in Parkinson’s disease (PD) patients, where direct stimulation of the subthalamic nucleus and *globus pallidus* was found to reduce rigidity, bradykinesia and iatrogenic dyskinesias [3,4]. The application of this technique is limited to early-to-mild cases when the nigrostrial pathway has some activity. However, this technique has limitations since the use of DBS (six weeks on/off) was not associated with improvements in cognition for cases of PD-derived dementia [5].

Cell therapy can provide a reliable methodology for substituting degenerated neurons with healthy ones, aiming to restore the normal function of the brain. Stem cells have been widely studied for this purpose and several clinical studies using embryonic stem cells (ESCs), induced pluripotent stem cells (iPSCs) [6] mesenchymal/stromal stem cells (MSCs) [7] and neural stem cells (NSCs), are underway [8]. The application of ESCs or autologous stem cells for cell therapy is compromised for both ethical and practical reasons. However, the discovery of a method to obtain, from adult somatic cells, patient-specific iPSCs from which NSC can then be obtained and maintained [9] opens the possibility for personalized NSC or trophic-factor therapies [10]. The use of differentiated/somatic cells is also possible, as reported by Ma and colleagues in their clinical trial [8]. The results obtained after a four-year follow-up of directly transplanted fetal dopaminergic cells revealed improved motor function in the patients. Transplant was also deemed successful based on the increased concentration of fluciclovine F 18 (18F-DOPA) in the engrafted area.

The next generation of devices for treating neurological diseases should harness the advantages of both stem cell therapy and DBS for a potential cure. For this, a multidisciplinary approach is necessary and knowledge from diverse areas, such as biomaterials, nanotechnology, materials science, neuroscience, and stem cell biology, must be wisely used. In this review, we propose to evaluate the physical and biological fundamentals necessary for developing therapeutic strategies that rely on electrical stimulation. Although several reviews on electrical stimulation, materials science, and neurological diseases exist, none of them approaches this topic in a wholistic and integrated manner. Here, we propose to evaluate: (1) the biological and physical fundamentals behind the generation of electrical fields by living cells; (2) the consequences of manipulating cell bioelectricity using externally applied electrical stimulation; (3) the development of platforms/scaffolds capable of directing electrical stimulation to the cells; and (4) the benefits of using electroconductive polymers in their design. Poly(aniline) doped with camphorsulfonic acid (PANI:CSA) is used as our model of an electroconductive polymer. Therefore, we also discuss (Section 3.1) the physico-chemical determinants for the correct PANI:CSA manipulation in the efficient manufacture of platforms/scaffolds, and provide examples of PANI:CSA-based platforms used for the electrical stimulation of neural cells (Section 3.2). With this broad and multidisciplinary review, we hope to provide other researchers a steppingstone for the development of future therapies and more reliable in vitro models for studying neurological disease progression and drug screening. Although neural cells are the focus of this review, including neural cell bioelectricity and basic neural cell biology/biochemistry, we also consider other off-topic case studies, when deemed necessary, to better understand the manipulation of bioelectricity and the design of electrical stimulation platforms (e.g., eye/retinal epithelial cell bioelectricity).

## 2. Bioelectricity and Electrical Stimulation

### 2.1. Bioelectricity in Living Organisms

Natural electric fields exist in all living cells. In fact, their existence is evidence of the functional capacity of cells to regulate their membrane potential through the transport of ions and other charged molecules across it. For this, cells make use of ion channels, mainly for fast adjustments, and transporter proteins, for slow and prolonged maintenance of membrane potential.

Bioelectricity is a physiological phenomenon that exists in all living cells and influences their physiology. Bacteria are also sensitive to electric fields. They can propagate potassium waves to communicate with each other when encapsulated in biofilms [11]. Moreover, certain anaerobic bacteria can control electron currents for nutrition and interaction with their environment, such as *Geobacter sulfurreducens* [12] and ones from the *Desulfobulbaceae* family [13].

Bioelectricity and regeneration in multicellular organisms are deeply connected. Cell-generated electricity can regulate morphogenesis in all living tissues. The regeneration of an organ involves careful regulation of cell division, movement, and positioning. It has been even proposed that, during embryogenesis, electrical cues serve to direct organ and limb formation. Moreover, bioelectrical fields can encode information regarding morphology and the patterning of tissue. For example, morphogenesis can be influenced by manipulating the membrane potential. The use of chemical cocktails that either depolarize or hyperpolarize the cell membranes of severed areas in planaria can induce either the formation of a new head or tail, respectively [11,14]. Bioelectricity is also important for the regeneration of limbs and lost organs in certain animals, such as frogs, lizards, and axolotls. Ion channels have been identified as important for tail regeneration in lizards [15].

Bioelectricity and genetic pathways are also interconnected in neural cells. The expression of ion channels (e.g., chloride channels) or transporters regulates the activity membrane’s electrochemical potential and can be signaled by the cell [14,16]. Conversely, changes in the membrane’s electrochemical potential can be induced by these proteins, triggering genetic responses for cell behavior changes and adaptation. In neurons, for example, the presence or lack of stimulation can determine cell survival and synapse formation or cell death [17]. Also, bioelectrical signals are proposed to encode the positioning and polarity of cells and are described as being critical in the regulation of many physiological phenomena, such as wound healing, neural-circuit shaping and body polarity in planarias, axolotls and frogs [11,18]. Therefore, we expect that the manipulation of bioelectric fields can have positive consequences for regenerative medicine. This is due to the importance of bioelectricity for different cells and organisms, as it is involved not only in homeostasis but also in morphogenesis. However, what are the general cell mechanisms involved in bioelectricity generation?

### 2.2. The Role of Membrane Potential on Bioelectricity

Bioelectricity is the result of the generation of electrical current by living cells and/or organisms. Bioelectricity depends solely on living cells, as it arises from gradients of charged species established across the intracellular and extracellular faces of their membrane (transmembrane electrochemical/voltage potential), a process intimately dependent on adenosine tri-phosphate (ATP) use. It starts with the necessity of cells to keep diverse ionic concentrations in check for the maintenance of cellular metabolism. First, cells express membrane transporters that either actively (ATP-dependent) or passively (ATP-independent) keep sodium, potassium, chloride, hydrogenocarbonate and hydronium ion gradients across the cellular membrane. This, in turn, allows cells to transport important nutrients intracellularly, such as glycose and aminoacids, and allows for the release of acidic metabolism byproducts, such as lactate, to the extracellular milieu [19,20]. The rapid manipulation of ionic concentrations and transmembrane electrochemical potential in cells can also be done actively by cells using ion channels. These are transmembrane proteins that control the rapid flow of ions between cells and/or between cells and the extracellular matrix (ECM), being sensitive to ligands (sodium channels) or membrane voltage changes (e.g., calcium channels). Cells that use these quick-response elements include nerve and muscle cells [21].

The eye’s lens is an example of a well-studied organ with homeostasis and regeneration dependent on bioelectricity. The lens enables the passage of light into the retina, and, with the help of the ciliary muscles, is also responsible for focusing this light. Epithelial cells on the lens undergo migration and differentiation to generate the transparent structures in the eyes known as the lens. The differentiation process involves, in the first stage, the synthesis of lens-specific proteins (e.g., aquaporins, α- and β-crystalin and beaded filament structural protein 2 (BFSP2)), the elongation and orientation of epithelial cells, followed by the destruction of every cellular organelle present (including the nucleus), and finally the packing and water exclusion from the formed core. Cao and colleagues [22] hypothesized that this process is orchestrated naturally by a cell’s own membrane potential. In fact, lens cells at a terminal-differentiation stage present a hyperpolarized membrane, while less mature epithelial and fibrous cells have a more depolarized membrane (difference of 32.5 ± 1.8 mV). This process is dependent of Na^+^/K^+^ ATPase activity, with higher expression in mature denucleated cells. The authors hypothesize that this naturally generated electric field is the main trigger for the differentiation of cells. Bioelectricity and genetic pathways are interconnected in a dynamic system. The expression of ion channels or transporters regulates the membrane’s electrochemical potential and can be signaled by the cell. This occurs in neurons, which can increase the expression of chloride channels to increase their stimulation threshold in response to constant stimulation [15,16]. Conversely, changes in the membrane’s electrochemical potential can be induced by these proteins, triggering genetic responses for cell behavior changes and adaptation. In addition to determining cell death/survival [17], bioelectrical signals are proposed to encode the positioning and polarity of neural cells and are described as being critical to the regulation of many physiological phenomena, such as neural development, wound healing, and neural-circuit shaping [11,18].

When multiple cells intimately interact to form specific domains or tissues, they establish more direct forms of communication. This is done by gap junctions, composed of channel proteins (connexins) that directly connect cell membranes. Using these gap junctions, cells can distribute and stabilize membrane electrochemical potentials between them, rapidly transmit bioelectrical changes (ionic currents), and coordinate cell responses and cell patterning. These are also known as electrical synapses, in opposition to conventional neural–neural synapses, due to the direct flow of ions/charged species between cells. In neural tissue, their main role is to synchronize cell activity [23,24]. For example, a recent study by Liu and colleagues illustrates that, in *C. elegans*, gap junctions can enhance synaptic transmission between interneurons and motor neurons [25].

Membrane potential is widely used by cells to exchange signals, especially through gap junctions and synaptic connections. Epithelial cells, for example, are known to establish two different membrane potentials across their apical and basal membranes (trans-epithelial potential—TEP), that are otherwise kept stable throughout a cell’s life. When the cell layer is damaged, through wounding or the mechanical removal of some of the cells, there is a short circuit and a steady and long-lasting electric field of low-intensity (140 mV mm^−1^) results. However, this field is strong enough to induce cell proliferation and migration to the lesion area. Once repairs are done and the cells re-establish the continuous cell layer, the electrical field is interrupted, making the cells return to their basal state. The same process occurs in damaged nerves. Overall, in the human body, the magnitudes of electric fields relevant for regeneration are between 1 and 100 mV mm^−1^, 10-times less than what is necessary for the stimulation of neurons (1–2 V mm^−1^) and 1000-times less than what is necessary for electroporation (100–500 V mm^−1^) [26].

The transmembrane electrochemical potential in cells is dynamic and changes with metabolism, cell cycles, and differentiation states. This is true even between different types of stem cells, such as, iPSCs, human embryonic stem cells (hESCs) and MSCs. For example, although both iPSCs and hESCs are electrophysiologically stable, their potassium-channel transcriptomes are slightly different. Regarding overall cell-population membrane potential and electrophysiological activity, MSCs present great heterogeneity whereas iPSCs (obtained from fibroblasts) and hESCs are more homogeneous [27,28,29]. These differences also indicate differences in the metabolism and even differentiation potential of MSCs within the same pool, while for iPSCs and ESCs, this does not occur. The transmembrane potential of fully differentiated cells varies according to cell type. Differentiated cells, such as neurons, have more hyperpolarized (more negative) resting membrane potentials (−70 mV). In contrast, cells with higher proliferative potential (cancer cells, MSCs, hESCs) have higher (more positive) resting potential values [14].

### 2.3. Direct Effect of Electrical Stimulation on the Cells

The manipulation of the bioelectricity of cells and tissues is made possible by the use of externally applied electric fields. According to McCaig and colleagues [26], in the human body, the electric fields relevant for regeneration are in the range 1–100 mV mm^−1^, which are 10-fold lower than the fields needed for stimulating neurons (1–2 V mm^−1^) and 1000-fold lower than the field necessary for electroporation (100–500 V mm^−1^). Most cell experiments are carried out with such field values, and the best value is usually chosen when visible changes in morphology (stretching and alignment) are observed.

Electrical stimulation protocols are usually optimized for each cell type and some of the optimized parameters include: (a) the stimulation regimen (e.g., direct current (DC), alternate current (AC), pulsatile direct current (pDC)) [30] (b) the respective signal frequency (e.g., 100 Hz) [31,32] (c) the electric field intensity (e.g., 1 V cm^−1^) or applied current (e.g., 100 µA) [33,34] (d) the duration of the stimulation (e.g., 12, 24 or 48 h) [30,35] and (e) the phase of cell development at which the electrical stimulation is applied (e.g., proliferation and/or differentiation stage) [36,37,38]. Depending on the parameters used, the manipulation of bioelectrical fields can greatly affect cell homeostasis. This is true for both matured neural cells and stem cells. Current studies in the literature show that externally applied electrical stimulation can impact neural stem cell morphology, migration, proliferation, apoptosis, and differentiation.

Neural cells and neurites align when stimulated by an electric field, a common hallmark. However, there is no consensus concerning the optimal orientation and the importance of this orientation, as they can both align parallelly or perpendicularly to the direction of the electric field. The adoption of the perpendicular orientation has been attributed to a minimization of the induced voltage drop across the cell body. This was observed by Chen and colleagues [39] for PC12 cells grown on bacterial cellulose (BC)-poly(3,4-ethylenedioxythiophene) (PEDOT) microfibers. However, these authors did not consider the influence of topography on cell alignment. In fact, Koppes and colleagues [40] found that substrate topography motivation orientation, provided by electrospun microfibers (2 µm in diameter), is the main determinant factor for cell and neurite alignment of dorsal root ganglia (DRG) neurons. They found that electrical stimulation (DC, 1 mA, 0.5 V cm^−1^, 8 h) only enhanced the length of the obtained neurites, while the direction was dependent on fiber orientation. Similar observations for the effect of combined topography and electrical stimulation were reported by other groups [38,41,42]. In another work, Koppes and colleagues [43] also observed specific changes on neurites. They observed that the electrical stimulation of dorsal root ganglion (DRG) neurons (DC, 0.1 mA, 0.1–1 V cm^−1^, 8 h) enhanced neurite extension and alignment with the electrical field. Although similar neurite extensions were observed when DRG neurons were co-cultured with Schwann cells, neurite alignment did not occur. Overall, we can conclude, based on the gathered results, that cell and neurite alignment is determined by substrate topography, even when electrical stimulation is applied. However, when the influence of topography is minimized, cell and neurite alignment will depend on the direction of the electric field [37,42].

Sun and colleagues [44] explored the impact of electrical stimulation on co-cultures of DRG neurons and Schwann cells on graphene-oxide (GO):PEDOT films. Electrical stimulation was found to significantly increase neurite length when voltages of 15 and 30 mV were applied but decreased when 60 mV was applied (with a distance between electrodes of approximately 1 cm). Moreover, at 30 mV, they found a higher number of Schwann cells interacting with the neuronal axons, even at large distances (such as 1506 µm from the central aggregate), and the formation of higher number of synapses. This suggests that an increased myelination of the cultured neurons occurs with electrical stimulation. The controlled release of the TrkB agonist 7,8-dihydroxyflavone, a neuroprotective drug [45] led to the potentiation of the above-described effects of the electrical stimulation.

Electrical stimulation can promote cell proliferation. Cell-cycle progression appears to be intimately dependent on membrane potential. In frog embryos, depolarized cells can signal distant cells in the neurocrest to proliferate [14]. Similarly, only depolarized astrocytes can proliferate and migrate in response to injuries [46]. Ghasemi-Mobarakeh and colleagues [47] conducted a proliferation study with NSCs where a short electrical stimulation (1 V cm^−1^) applied for 60 min was associated with an increase in cell number and higher neurite extension (30 µm for stimulated vs. 22 µm for control). Shorter stimulation periods (15 and 30 min) did not produce the same effect. Similar observations were made by Xu and colleagues [48]. When human neural stem cells (hNSCs) were electrically stimulated for seven days, they proliferated faster than their non-stimulated counterparts. Similar to the results of Ghasemi-Mobarakeh and colleagues, Xu and colleagues increasing voltages (15, 35 and 75 mV) induced an overall decrease in cell proliferation and an increase in neurite length.

Electrical stimulation can also enhance the expression and release of bioactive molecules by proliferating NSCs. Song and colleagues [49] observed an increase in the mRNA expression of brain-derived neurotrophic factor (BDNF), glial cell-line-derived neurotrophic factor (GDNF) and neurotrophin 3 (NT3) after one hour of electrical stimulation and 24 h of resting using a 2D experiment, and this effect was enhanced when cells were electrically stimulated in a 3D experiment. Other markers, such as heparin-binding epidermal growth factor-like growth factor (HBEGF) (important for cell growth and glia-derived progenitor formation) and heat shock protein family member 1 (HSPB1) (important as a protein chaperon, neurofilament homeostasis and axonal transport), were also upregulated after electrical stimulation, whereas vascular endothelium growth factor A (VEGF-A) (pro-angiogenic factor) expression did not change, and enolase 2 (ENO2) (associated with neural lineage in NSCs) expression decreased.

Electricity can improve the differentiation yield of neural stem cells. Yang and colleagues [42] showed that the highest expression of microtubule-associated protein 2 (*Map2*) and class III beta-tubulin (transcript) (*Tubb3*) occurred with electrical stimulation regardless of the topography used. Interestingly, the combination of electrical stimulation with topography allowed for the differentiation of neural cells capable of generating sodium current, with the highest expression of voltage-gated sodium channel subunit 1 alpha (*Scn1α*) and calcium voltage-gated channel subunit 1C (*Cacna1c*). The authors noted that electrical stimulation did not interfere with the expression of glial fibrillary acidic protein (*Gfap*), suggesting that astrocytic differentiation was not promoted. Finally, since the expression of octamer-binding transcription factor 4 (*Oct4*), homebox protein nanog (*Nanog*) and Nestin (*Nes*) was not impacted by electrical stimulation, the protocol used was considered efficient for cell differentiation. In fact, altering the cell membrane potential may not only interfere but also direct the fate of the differentiation of stem cells into specific cell types [15]. The positive effects of electricity on specific neural gene expressions in differentiating cells are also described in another work [48].

Other studies have also explored the impact of electrical stimulation on neural cell differentiation. Pires and colleagues [32] previously showed that the application of an electrical field (1 V cm^−1^) to neural stem cells on cross-linked PEDOT doped with poly(styrene sulfonic acid) (PEDOT:PSS) films promoted elongation of the cells and increased neurite length and neuron-specific class III beta-tubulin (protein) (TUJ1) expression. Zhu and colleagues [33] also demonstrated increased TUJ1 and MAP2 expression and increased neurite length when cells were grown on cross-linked PAN electrospun fibers. Borah and colleagues [50] observed increasing neurite numbers and length when an electric field was applied to PC 12 pheochromocytoma cells on coaxial poly [2-methoxy-5-(2′-ethylhexyloxy)-1,4-phenylene vinylene] (MEH-PPV): poly(ε-caprolactone) (PCL) electrospun fibers. Finally, Koppes and colleagues [40] tested different electric field strengths (10, 50 and 100 mV mm^−1^) for rat dorsal root ganglion neurons (DRG) cultured alone or in co-culture with Schwann cells. They found that, in the case of DRG neurons cultured alone, neurite outgrowth increased from 700 µm to 1260 µm when they applied an electric field of 50 mV mm^−1^. Neurite length further increased to 2229 µm when DRG neurons were co-cultured with Schwann cells and subjected to an electric field of 50 mV mm^−1^. The authors argued that the nerve/neurotrophic growth factor (NGF) secreted by the Schwann cells was responsible for this boost.

Electrical stimulation can also influence the neural lineage of differentiated NSCs. Yang and colleagues [42] indicated that the combination of nanotopography and an electroconductive surface can guide hNSC differentiation into inhibitory neurons (gamma-aminobutyric acid (*Gaba*)+) instead of excitatory neurons (vesicular glutamate transporter 1 (*Vglut1*)+). In the work of our group, Garrudo and colleagues [38] reported a similar trend for neural differentiating hiNPC. However, when electrical stimulation was applied (pDC, 1 V cm^−1^, 100 Hz, 12 h per day, 30 days) excitatory neuron (VGLUT1) and maturation (neurofilament heavy polypeptide (*Nef-H*), synaptophysin (*Syp*), CACNA1C, SCN1α) markers are overexpressed, and early maturation (neuron-specific class III beta-tubulin (transcript) (*Tubb3*), doublecortin (*Dcx*), tubulin-associated unit (*Tau*)) and inhibitory neuron (glutamate decarboxylase 67 (*Gad67*), vesicular GABA transporter (*Vgat*)) markers were downregulated. Interestingly, scanning electron microscopy (SEM) images suggest that the axon diameter dramatically increases with electrical stimulation, which coincides with the upregulation of NEF-H. VGLUT1 fluorescence intensity, an indirect measurement of protein content, was also increased, which coincides with the upregulation of VGLUT1 (transcript).

The bioelectricity of cells, and their response to electrical stimulation, can also be modulated pharmacologically by using substances that interfere with: (1) connexins; (2) ionic channels; or (3) the release of neurotransmitters, which causes changes in the cells’ transmembrane electric field and can lead to similar effects in gene expression. However, due to the ubiquity of these proteins and the number of different functions they regulate in different cells in the human body, these effects might be more unspecific. As such, there is a need to further study the biomolecular mechanisms behind the signal transduction of electrical stimulation and its effects.

#### Biochemical Pathways Involved in Signal Transduction of Electrical Stimulation

The effects of electrical stimulation on neural cells are well documented, and the literature will surely expand on this subject. However, the molecular mechanisms involved in the signal transduction from an electric field to the activation of specific/multiple molecular cascades inside the cells are yet to be conclusively established.

The current literature provides evidence that voltage-dependent calcium channels are the starting point for neural cell response to externally applied electric fields. This is evidenced by numerous reports in the literature where calcium levels increased in cells subjected to electrical stimulation when cultured on electroconductive substrates [39]. Calcium channels play an important role in the electric-chemical signal transduction in neurons and are important for both cell communication and adaptative responses. Voltage-dependent calcium channels enable extracellular calcium influx into the intracellular environment and lead to the release of synaptic vesicles containing neurotransmitters into the synaptic cleft. Intracellular calcium can also act as a secondary neurotransmitter, and, through calmodulin/calcineurin binding, can lead to the activation of proteins involved in the mitogen-activated protein kinase (MAPK)/extracellular signal-regulated kinase (ERK) pathway (Ras, MAPK, ERK1/2) and Ca^2+^/calmodulin-dependent protein kinase (CAMK)-IV, leading to the consequent activation of the transcription factor cAMP-response element binding protein (CREB) [51,52]. This ultimately leads to the increased expression of several neural specific genes. It is also possible that voltage-dependent calcium channel activation occurs due to conformational changes, which can lead to an increased influx of calcium to the neurons.

Another potential mechanism that can explain the transduction of electrical stimulation into changes in neural cells is direct membrane depolarization. Cao and colleagues [22] observed such a phenomenon in differentiating human lens epithelial cells (HLECs). In a normal lens, the bioelectric field is established by the selective Na^+^/K^+^ transporter activity and leads to different intensities depending on the area and degree of differentiation of the cells. Although not directly quantified, the difference between the electric fields (V_mem_) of the differentiating fiber zone and the fully differentiated fibers was estimated to be 32.5 mV, which was reduced to around 11 mV when ouabain (Na^+^/K^+^ transporter inhibitor) was used. The stimulation of cultured HLECs through the application of 1 V cm^−1^ on the culture medium significantly increased their differentiation. This was demonstrated by increasing intracellular amounts of phosphorylated forms (activated) of protein kinase B (Akt) (survival pathway) and cyclin-dependent kinase 2 (CDC2) (nuclear assembly during differentiation), and a rapid increase in β-crystallin and aquaporin expression. The authors attributed this effect to direct membrane depolarization of immature fiber HLECs, possibly caused by ion movement in the culture medium. Nevertheless, a lack of a pharmacological control and the presence of regular and voltage-dependent ion channels in HLECs limits the generalization of these conclusions [53,54].

Calcium channel downstream cascade activation and membrane depolarization appear to be opposing mechanisms to explain the improvement of differentiation in NSCs. However, these mechanisms might be intimately associated with, and may influence or lead to, the activation of each other. Wang and colleagues [55] describe one of the first pharmacological studies on the influence of selective voltage-dependent channel inhibition during NSC differentiation on a 3D scaffold. Three important conclusions were made: (1) the application of electrical stimulation enhanced NSC differentiation; (2) the selective inhibition of sodium (amiloride hydrochloride), potassium (4-aminopyridine), calcium (nifedipine) and chloride (4,4′-diisothiocyanostillbene-2,2′-disulfonic acid) voltage-dependent channels affected NSC normal differentiation; and (3) the NSC normal differentiation was rescued by the simultaneous application of electrical stimulation in all four treatments. The phenotype rescued with electrical stimulation was more pronounced on amiloride- and nifedipine-treated cells, but no statistically significant differences between the two conditions were reported.

Other possible mechanisms involve changes in cells biomechanics, including: (a) the reorganization of cell receptors (e.g., clustering of BDNF receptor or integrins) and consequently; (b) of the cytoskeleton; (c) modulation of the activity of other ion channels; (d) interference with mitochondrial metabolism; and (e) epigenetic changes [56]. However, these can also be a consequence of the mechanisms described before and may vary depending on the cell type and maturation stage. As such, there is still no direct evidence of a common transduction mechanism for the effect of electrical stimulation on neural cells, and more research needs to be conducted.

### 2.4. Methods for Electrical Stimulation—The Importance of Support Materials

Bioreactors need to be designed to perform the electrical stimulation of cells, as regular cell-culture materials are not designed for this. Both the culture medium and the stimulation process must coexist without either compromising each other’s quality or negatively impacting the scaffold or the cultured cells. Different types of bioreactors are possible depending on the element that is directly subjected to electrical stimulation: (a) the culture medium; or (b) the supporting material (Figure 1) [57,58]. Scaffolds for neural tissue engineering can be stimulated with both types of designs.

Culture media bioreactors (Figure 1A,(A1)) rely on the application of an electrical field to the culture media in which the cells are maintained. Ions and charged molecules are forced to migrate to either of the electrodes used depending on their charge. Once the stimulus is removed, the homogeneous distribution of charged molecules resumes. The only limitation of this method is the potential hydrolysis of water when the applied potential rises above 1.23 V, promoting the formation of charged chemical species such as OH^−^ and radicals OH^•^ (faradic products) [59,60].

We consider supporting material bioreactors (Figure 1B,(B1)) as those which have an electroconductive substrate that short-circuits electrons between the used electrodes. Typically, cells are cultured on top of the electroconductive material and are directly influenced by it. Since water hydrolysis is minimized, higher voltages can be applied to cells cultured directly over them. The electroconductivity of the support material is important in order to maximize current passing through the material or to minimize the applied voltage. Other key properties for the support material used include: (1) electrochemical properties; (2) structure; (3) stability in aqueous solutions; and (4) biocompatibility. Some examples of reported bioreactors are summarized in Table 1.

The applied electric field distribution on an electroconductive material should be homogeneous throughout the sample length, independently of its structure. This thematic was explored by Song and colleagues [49], using finite element simulation to evaluate the distribution of the applied electric field on a 2D poly(pyrrole) (PPY) film versus a 3D PPY-coated hydrogel. They observed that the effective electric field was evenly distributed through the PPY film/coating (0.536 ± 0.095 V cm^−1^ and 0.511 ± 0.025 V cm^−1^ for 2D and 3D experiments, respectively). The only exception was the area of the contact points with the electrodes where the electric field was twice the value (0.9 V cm^−1^). For the 3D construct, the authors did not consider the influence of the ionic conductivity of the hydrogel used for cell encapsulation on the penetration of the electrical field. As such, no conclusions could be made about the distribution of the electric field inside the hydrogel and whether seeded cells could be influenced by it. Meneses and colleagues [61] modeled the spatial pattern of electric fields established in the culture medium surrounding a 3D scaffold with different electroconductivities, following electrical stimulation. Scaffolds with lower electroconductivities, including poly(lactide acid) (PLA) (1 × 10^−12^ S cm^−1^, not electroconductive) and chitosan–graphene (2.5 × 10^−3^ S cm^−1^, mild electroconductive) were associated with high culture-medium polarization (5–6 V m^−1^), whereas titanium scaffolds with higher electroconductivity (7.5 × 10^3^ S cm^−1^, highly electroconductive) were not. Such observations can be attributed to a dominant electron conduction with increasing electroconductivity of the material used.

The production of pure electroconductive scaffolds is mostly confined to 2D films due to the conductive polymer’s limited processability. Even so, research has shown positive results in the use of electroconductive scaffolds on cell growth and differentiation, especially those studies focused on the initial biocompatibility assessment of new materials or when studying the isolated effects of electroconductivity and/or electrical stimulation on cells [42,49]. Alternatively, the processability of electroconductive polymers into 3D structures can be enhanced by: (1) direct coating/electrodeposition on pre-made 3D structures [62,63]; or (2) blending with carrier polymers, allowing the production of many diverse structures, such as electrospun fibers [64], extruded 3D structures [65,66] and hydrogels [67,68]. Interestingly, the choice between a 2D and a 3D structure is not trivial and carries different consequences on the cell phenotype obtained. For example, Song and colleagues [49] studied the effect of a 2D vs. a 3D structure of PPY films on the phenotype of human-induced neural progenitor cells (hiNPCs) (encapsulated in alginate hydrogels) under electrical stimulation. Interestingly, after just one hour of electrical stimulation (0.4 V cm^−1^), the authors observed an increased expression of HBEGF, correlated with an increased formation of glial precursor cells, HSPB1, associated with the stress response from cells after electrical stimulation, and NT3, a neurotrophin responsible for neuron maturation, for both the 2D and 3D structures. However, cells increased the expression of BDNF and GDNF (neurotrophic factors responsible for neurite elongation and neuron maturation) only when growing in the 3D structure. This effect might be the consequence of a more evenly distributed electrical field through the 3D structure, as predicted by the finite element analysis. Scaffold limitations, such as low levels of oxygen/glucose inside the hydrogel, should also be considered as possible explanations for the observed secretion of neurotrophic factors. However, electrical stimulation dramatically improved neurotrophic-factor secretion on both 2D and 3D samples. This also justifies the use of more complex and intricated tissue-mimicking structures as reliable platforms for electrical stimulation.

**Table 1 polymers-15-02674-t001:** Electrical stimulation devices used in the literature and relevant for neural cells: components, type of stimulation performed on neural cells and the observed outcomes.

Type	Cells Used	Stimulated Substrate	Types of Electrodes	Electrolyte Solution Used	Stimulation	Power Source Used	Signal Frequency (Hz)	Duration	Outcomes	References
Direct substrate stimulation	mNSCs	Cross-linked poly(acrylonitrile) (PAN) electrospun fibers	Platinum + printed circuit board	(not used)	100 µA (asymmetric biphasic)	asymmetric biphasic programmable electrical device and a printed circuit board	100	24 h, 1 day after seeding. Cells were allowed to differentiate for more 7 days.	When electrical stimulation was performed, it was observed at the end of the experiment: - Increased cell number; - Increased neurite length; - Increased *Tubb3* (qPCR) and *Map2*/MAP2 (qPCR and IF) and decreased expression of GFAP (IF).	Zhu and colleagues, 2017 [33]
Direct substrate stimulation	hNSC	Nanopatterned Titanium coated PUA	Copper wire + PDMS	(not used)	DC (pulsed), maximum 3 µA and 25 V,	programmable digital power supply MK3003P	1	30 min, 2 times a day, 5 days	Enhancement of hNSC differentiation, independently of the type of substrate used; Increased expression of TUJ1 and MAP2 on differentiated neurons	Yang and colleagues, 2017 [42]
Direct substrate stimulation	NSCs	PCL-PANI_Gelatin electrospun fibers	1 platinum and 1 silver electrodes	(not used)	DC (1 V cm^−1^)	(not referred)	(not referred)	15, 30 and 60 min.	Increased cell number 1, 3 and 5 days after stimulation. Longer neurite extent (30 µm vs. 22 µm)	Ghasemi-Mobarakeh and colleagues, 2009 [47]
Direct substrate stimulation	hNSC (ReN-VM), p3-5	PANI coated PVV hydrogel	PANI-coated Indium Tin Oxide	(not used)	AC—Charged-Balanced biphasic (15, 35 and 75 mV)	Agilent B2912A precision source/measure unit	200	Every 6 h for 1,3,5 and 7 days	Enhanced cell proliferation, that decreased for higher voltage values. Enhanced neurite extension with increasing voltage values. Enhanced neural gene expression with electrical stimulation	Xu and colleagues, 2016 [48]
Dual system of conductive stimulating layer and inert cell support	iNPCs	Alginate hydrogel with cells encapsulated on top of (2D) or surrounded by a PPY film (3D).	Silver wire + silver paste	(not used)	AC, 0.4 V cm^−1^	(See Oh 2018)	100 Hz	1 h of stimulation + 24 h of resting period	Electrical stimulation enhanced the mRNA expression of HBEGF, HSPB1 and the neurotrophins BDNF, GDNF and NT3. 3D structure + electrical stimulation boosted mRNA expression of BDNF and GDNF	Song and colleagues, 2019 [49]
Direct substrate stimulation	mNSCs	PPY-coated PAN (no cross-linking) electrospun fibers	Stainless Steel	(not used)	100 mV cm^−1^,	AFG3022C, Tektronix, USA	100	4 h of stimulation, total of 7 days.	- Enhanced cell maturation through increased Tau protein expression; - Prevention of neurons growing and differentiating into clusters - Enhanced proliferation of glial cells	Xu and colleagues, 2018 [69]
Direct substrate stimulation	PC12	PLCA-SF-PANI electrospun fibers, monoaxial and hollow co-axial	Similar to Ghasemi-Mobarakeh L et al. 2009	(not used)	(100 mV cm^−1^)	(not referred)	(not referred)	1 h per day, 5 days in total.	Increased neurite-positive cells and respective length.	Zhang and colleagues, 2014 [70]
Direct substrate stimulation	PC12	indium doped tin oxide (ITO) needle coated with PANI	“Wires”	(not used)	AC (100 µA)	TBSI Neural Stimulator ()V1.0.8, Triangle BioSystems, Durham, NC, USA	1Hz (1 s repeat interval)	1, 2 and 4 h + 24 h of resting	Cell density higher in the following order: 4 h = 2 h > 1 h > 0 h Increased neurite length with increased duration; Increased protein adsorption with electrical stimulation	Wang and colleagues, 2015 [71]
Stimulation of the culture media	Dissociated neurons from *Xenopus laevis*	Culture media (20% Liebowitz L15 culture medium, 2% penicillin/streptomycin, 1% fetal bovine serum) made in Steinberg’s solution	Ag/AgCl electrodes, indirectly connected to culture through agar bridges.	Steinberg solution for the electrode solution and to prepare the culture media.	DC—50–133 mV mm^−1^ (low field strength) and 143–200 mV mm^−1^ (high field strength)	(not referred)	(not referred)	2–4 h after seeding + 5 h of stimulation	Electrical field induced neurite orientation to the cathode; Addition of CS-6S rich GAGs to culture media enhanced neurite turning, whereas CS-4S rich ones inhibited;	Erskine and colleagues, 1997 [72]

#### The Importance of Electroconductive Materials

Scaffold development for tissue-engineering applications relies on the optimization of biomaterials and their nanostructuring to support NSC growth, differentiation, and maturation. Biomaterials can be viewed as engineered materials (natural, synthetic or hybrid) with bioactive properties, and their therapeutic and toxicologic profiles must be compatible in order to directly contact living systems. In fact, their acceptance both by cells and tissues depends on how their physical (2D/3D structure, electroconductivity, hydrophilicity, mechanical properties, porosity, zeta potential), chemical (chemical functionalization, co-polymerization, molecule immobilization) and pharmacological properties influence or enhance their blending/bioactivity capacity. Through the finetuning of these parameters, we can modulate cell behavior and mimic the target tissue microenvironment, enhancing the material’s biocompatibility and easing its clinical applicability to stem cell therapy [42,73,74]. Biocompatibility can be screened using both in vitro (cultured cell lines and stem cells) and in vivo (biological samples, animal models) assays, with experimental timespans depending on the effects under observation [74].

Electroconductive polymers were shown to be capable of sustaining cell growth [75] and conduct electrical stimuli to cultured cells. Electroconductive polymers are composed by a sequence of monomers that form a conjugated single–double bond system, that is, an alternation of π-bonds. Through doping, either by charge or proton transfer, it is possible to induce charged defects in this regular bond alternation that can be propagated through the polymer chains. This delocalization capacity can be quantified through the determination of a polymer’s electroconductivity (σ), usually expressed in S m^−1^ (SI) or S cm^−1^ (the most-used in scaffold design). These polymers are also particularly advantageous for other applications such as bioelectronics, due to their combined ionic/electronic conductivities, which decreases their interfacial impedance in biologic fluids, and improves the performance of signal transmission between sensors and cells/tissues [76,77]. When compared to metals, electroconductive polymers are easily processed into 3D structures at relatively low temperatures, have more favorable mechanical properties [32] and can be easy and cheap to produce [78]. These features justify the wide range of additive manufacturing techniques that can be employed in their processing and also make these ideal candidates for designing scaffolds for the potential therapy of neurological diseases [79,80]. Moreover, electroconductive polymers can slightly improve NSC differentiation even in the absence of electrical stimulation [42,81].

Some examples of electroconductive polymers include PANI, PPY and PEDOT [82]. All these polymers require doping agents to create and stabilize their most conductive form, by either charge transfer doping or proton transfer. While PANI is mostly doped via proton transfer, using (mild-) strong acids (e.g., CSA, hydrochloric acid, phosphoric acid, sulfuric acid), PEDOT is stabilized in the oxidized form during its chemical synthesis, using for instance hyaluronic acid or poly(styrene sulfonic acid) [83,84,85]. Usually, the doping agent’s proportion to the conductive polymer must be optimized to obtain the best electric conductivity values. After a certain value (usually 50–60%), electroconductivity starts to decrease with increasing amounts of doping agents.

## 3. PANI:CSA, a Versatile Electroconductive System Suitable for Neural Tissue Engineering

The use of PANI as an electroconductive substrate for electric stimulation of cells in biomaterials has already been reviewed by Arteshi and colleagues and Qazi and colleagues [74,86]. Moreover, more specialized reviews, such as that by Wei and colleagues [87] summarize the design of conductive fibers using PANI, and other conductive polymers, for diverse medical and soft-electronics applications. In this section, we intend to complement such reviews by exploring the physico-chemical determinants necessary to improve PANI:CSA electroconductivity and the design of platforms/devices for neural applications. We propose to explore these in more detail and explore their influence on the biological performance of platforms for the electrical stimulation of neural cells. We will also place a special focus on in vitro assays due to their wide availability in the literature, while complementing this focus with in vivo assays when possible.

PANI is regarded as a good electroconductive and inexpensive material candidate for the design of electroconductive scaffolds for tissue engineering [88]. When doped, PANI undergoes a change in its resonance structure, switching from the non-conductive emeraldine base (blue) form to the highly conductive emeraldine salt (green) form. This stable salt is the most conductive form of PANI [89,90]. PANI can be doped with several doping agents with positive effects on the polymer’s electroconductivity, including CSA, formic acid, hydrochloric acid, picric acid, and phosphoric acid [91,92,93]. CSA is the most widely used doping agent for PANI (Figure 2), and an optimal proportion between the two is necessary to obtain the highest conductivity value. Such optimization studies were already performed by Holland and colleagues. The authors tested different doping levels of PANI:CSA (10–90%) and concluded that a doping level of 60% resulted in the highest electroconductivity at RT with metallic behavior, even at low temperatures [88].

### 3.1. PANI:CSA–Processing Methods, Solvent Systems and Stability

The molecular organization and electroconductivity of PANI:CSA are influenced by the solvent(s) used in its processing, such as *m*-cresol, hexafluoropropanol (HFP) and trifluoroethanol (TFE). These consequences are described in terms of a pseudo-doping effect of the solvents used. A pseudo-doping agent is capable of inducing permanent structural and electroconductive changes to PANI:CSA, in combination with, or even without, a primary doping agent (e.g., CSA). It chemically interacts with PANI:CSA and promote the uncoiling of PANI:CSA aggregates that are naturally formed when dispersed [94,95]. The positive consequences of pseudo-doping on the electroconductivity of PANI:CSA can be observed in numerous studies. Cao and colleagues [96] studied the effect of different organic solvents on the electroconductivity of cast PANI:CSA films, using as reference the conductivity of pressed films (10 S cm^−1^). They found that solvents such as *m*-cresol were able to solvate PANI:CSA (300 S cm^−1^), stabilizing it in a more rigid conformation, leading to an increase in conductivity. According to the authors, this arises from the hydrogen-bonding ability of *m*-cresol, allowing chemical interaction between the hydroxyl group of *m*-cresol and the amine groups of PANI. When hydrogen-bond acceptors such as dimethylsulfoxide (DMSO) or chloroform (CHCl_3_) were used, either alone or mixed with *m*-cresol, the conductivity decreased dramatically (10^−2^ S cm^−1^). The authors also describe the effect of other bulkier yet weaker hydrogen-bond donor solvents, such as 2-ethylphenol (228 S cm^−1^), trifluoroacetic acid (100 S cm^−1^) and hexafluoropropanol (60 S cm^−1^), which had a similar, but less pronounced effect, when compared to *m*-cresol. In a different work by Hopkins and colleagues [91], HFP was found to pseudo-dope PANI, allowing it to adopt an expanded conformation that increased electroconductivity and decreased chain aggregation. In contrast to *m*-cresol, HFP could be completely removed from PANI:CSA without compromising its electroconductivity. Fryczkowski and colleagues [97] also describe the positive effect of TFE on PANI:CSA electroconductivity when used for processing.

Changes in electroconductivity induced by pseudo-doping agents are concomitant to structural changes on PANI:CSA, which are readily visible through different spectroscopy/diffraction methods. Xia and colleagues [98] reported that processing PANI:CSA with *m*-cresol yields changes in the respective ultraviolet-visible (UV/Vis) and near-infrared (NIR) spectra, including: (1) reduced intensity of the band at 350 nm (π-π* transition); (2) increased intensity of the band at 440 nm (polaron-π* transition); (3) reduced intensity of the band at 780 nm (π-polaron transition); and (4) the formation of a wide and intense free-carrier-tail/conduction band in the region of 1000–2600 nm. These changes are explained by an expansion of the coil-like conformation of PANI:CSA aggregates, which improves electroconductivity by an increase in interchain contacts and the spatial extension of the conjugated chain segments. Similar results were observed by other works in the literature [99,100]. Yao and colleagues [99] describe an increased crystallinity of PANI:CSA when processed using solvent systems with increasing concentrations of *m*-cresol. Such effect was attributed to *m*-cresol inducing the formation of more frequent ordered regions in PANI:CSA samples, as suggested by increasing intensity of x-ray diffraction (XRD) peaks at 2θ = 20° and 25°. Such change is compatible with the observations of Lee and colleagues [101] which associate these changes to an increase in π-π stacking of PANI:CSA chains and a more planar and rigid chain conformation. Similar observations were also made by Chaudari and colleagues [92] for acid-doped PANI and by Hopkins and colleagues [91] for HFP-processed PANI:CSA. A planar and more rigid conformation of PANI:CSA has positive consequences for both electroconductivity, ionic conductivity and capacitance. This justifies the use of such systems in the design of supercapacitors [102,103] and presents an opportunity for the design of devices suitable for brain (cell) stimulation. Otherwise, electroconductivity will remain low even when high amounts of PANI are used to fabricate the films [104].

PANI:CSA properties can change with environmental conditions. PANI:CSA has a polyelectrolyte structure, with a high ability to absorb water. In the short term, this is beneficial for the material’s electroconductivity by favoring proton mobility through the polymer chain. In the long term, this effect promotes de-doping of PANI:CSA, compromising its electroconductivity [89,105]. An example of this can be found in the work of Hobaica and colleagues [106], where it was observed that the doping extension of PANI with hydrochloric acid decreased when samples were immersed in water at a pH of 1.18 for approximately 140 h. This led to a decrease in the material’s electroconductivity (6.25 to 1.94 S cm^−1^), which was not observed with prolonged storage in air (five years at room temperature). A similar effect was observed by Bidez and colleagues [107], whose PANI film’s resistivity increased by 1,000 times when incubated in Dulbecco’s modified eagle’s media (DMEM) (37 °C and 5% CO_2_) for 100 h. The best option available to overcome this limitation is the use of a carrier polymer to shield PANI from water contact. In fact, an improvement can occur, as was observed in the work of Qazi and colleagues [75], where the electroconductivity of PANI:CSA/poly(glycerol sebacate) (PGS) composites decreased by only 10 times (1.77 × 10^−2^ to 1.03 × 10^−3^ S cm^−1^ for PAN/PGS 30%) when immersed in phosphate-buffered saline (PBS) for four days. Another option is the design of coaxial electrospun fibers, where blended PANI:CSA is partly shielded from the environment, as we described in a previous study by our group [38].

The degradation profile of PANI was previously assessed. Using bidimensional spectrochemistry, Lopez-Palácios and colleagues [108] correlated the effect of the number of voltametric cycles with PANI weight loss. This was associated, in an initial stage, with the loss of synthesis byproducts entrapped in the polymers, and in a later stage with degradation and release of soluble products. This confirms the observations made by Kobayahi and colleagues [109] describing PANI’s oxidative degradation mechanism on electrodes. These results are important to fully understand the biocompatibility profile of PANI and the potential risks of using PANI in scaffolds for electrical stimulation.

### 3.2. Biocompatibility for Mammalian Cells

The biocompatibility of PANI has always been a subject of intense debate in the literature. Numerous in vitro and in vivo studies have demonstrated that PANI is biocompatible, either alone or when blended with other materials. For example, PANI is capable of the adsorption of proteins at its surface, avoiding its aggregation and making it a good substrate for cell adhesion and extension [62,71,110]. One of the most robust biocompatibility studies of PANI alone was performed by Humpolicek and colleagues [111]. In a first approach, biocompatibility tests following ISO-10993 were performed on PANI following several un-doping/re-doping cycles, showing that these operations had a positive effect on biocompatibility when human cancer cell lines (HepG2 and HaCaT) were used. Toxicity was therefore attributed to the presence of low-molecular-weight impurities arising from its synthesis. Interestingly, the base form of PANI was observed to be more biocompatible than its salt form [112]. A follow-up study by the same author [113] showed that, among all the possible impurities, ammonium persulfate, an oxidizing agent necessary for the synthesis of PANI, was found to be the most toxic species present. The toxicity of aniline monomers was also evaluated separately, and toxicity was observed above 0.25 mg mL^−1^ for aniline (An) and 0.75 mg mL^−1^ for aniline hydrochloride (AnH). When the monomers were combined with ammonium persulfate (as low as 0.1 mg mL^−1^) a synergistic toxic effect occurred for every combination of the concentrations tested. The biocompatibility of PANI doped with different acids was also analyzed. Its toxicity was again attributed to the presence of impurities. These results clearly indicate that PANI is a biocompatible polymer and that possible toxic effects on cells can be overcome by improving the material-purification steps after synthesis.

Stem cells are essential for building trustworthy in vitro disease models and tissue engineering strategies due to their ability to differentiate into mature cells. Since stem cells are extremely sensitive to the surrounding microenvironment, it is therefore paramount to determine their compatibility with poly(aniline) in the construction of conductive scaffolds. Initial work from Humpolicek and colleagues [112] showed that extracts from PANI emeraldine salt, but not emeraldine base, could impact the differentiation of R1 mouse embryonic stem cells into erythroid cells and cardiomyocytes (embryotoxicity). No impairments were observed when cardiomyocyte adhesion was evaluated on PANI films (base and salt forms). Likewise, with cell differentiation, both in cardiomyogenesis from mouse ESCs and neurogenesis from mouse neural progenitors, no negative effects were observed. In another study conducted by Xu and colleagues [48], coating poly(2-vinyl-4,6-diamino-1,3,5-triazine)-*co*-1-vinylimidazole) copolymer (PVV)-based hydrogels with PANI did not affect the material’s biocompatibility with hNSCs.

In vivo biocompatibility studies for PANI are also available in the literature. Zhang and colleagues [114,115] found that the oral LD50 (lethal dose of a substance that kills 50% of the animals in a test) of pure, in-lab-synthesized PANI nanospheres and nanofibers (40.9–100 mg mL^−1^) was higher than 3000 mg Kg^−1^. Interestingly, they also reported that, apart from the lethargy and weight loss observed when only PANI fibers were administered at concentrations higher than 64 mg mL^−1^, no other toxicity symptoms were observed. Kidney and liver pathohistological analysis revealed no changes in tissue architecture, except for the fatty degeneration of liver cells when mice were administered PANI fibers at 100 mg mL^−1^. The authors attribute this effect to the lower molecular weight of PANI fibers when compared with PANI nanospheres, which reportedly tend to aggregate and become toxic. Zhou and colleagues [116] showed that, also in mice, the intra-tumoral administration of colloidal PANI-poly(glutamic acid) nanogels was associated with normal tissue histology of heart, liver, spleen lung and kidneys 20 days post-injection. This suggests that leaking and circulating nanogel debris are safe. Regarding the local effects of implanted PANI, Qu and colleagues [117] observed that PANI limited inflammatory response and biodegradation after 28 days implantation of chitosan-PANI/dextran hydrogels in mice. Similar observations were made by Das and colleagues [118], whose silk fibroin (SF)-PANI electrospun nerve conduit enabled successful myelinization of damaged nerves in mice when transplanted for 12 months, especially when Schwann cells were co-transplanted, with no inflammatory response observed. Wang and colleagues [63] observed similar beneficial results in sciatic nerve regeneration of rats for their PANI-coated zein microtubes when transplanted for two months. The local inflammatory response was observed after four months due to PANI debris remaining in the tissue. All in all, PANI can be regarded as a biocompatible material for tissue-engineering applications, with both in vitro and in vivo data supporting this conclusion.

### 3.3. PANI:CSA and Blend Preparation for Scaffold Design

Due to its poor mechanical properties and inability to be processed in 3D structures alone, the use of PANI for tissue-engineering applications requires the use of carrier polymers. Different polymers can be used, including PCL, PGS, and PLCL. The most widely studied and promising scaffolds for neural tissue engineering include electrospun fibers [70,119] and hydrogels [120,121], where the authors make use of different PANI blends to fully take advantage of its electroconductive properties and positive effects on neural cells.

PGS is a polymer with important applications in neural tissue engineering due to its biocompatibility, biodegradability, and tunable mechanical properties. Conductive PGS composites were obtained by Qazi and colleagues [75] after blending with PANI:CSA (15–30%). The incorporation of PANI directly increased the system’s electroconductivity (maximum of ≈1.8 × 10^−2^ S cm^−1^), Young’s elastic modulus (0.32 MPa for pure PGS vs. 9.2 MPa for PANI:PGS 30%) and contact angle (86° for pure PGS vs. 107° for PANI:PGS 30%). In vitro stability/degradation of this scaffold was also studied and a constant decrease in 10-times the conductivity value was observed in just four days when incubated with PGS. In fact, the authors reported changes in the FTIR spectra of the composite correlated with the de-doping of PANI and degradation of the PANI:CSA:PGS composite after 15 and 45 days in PBS, which might explain these changes. Finally, the composite was proved to be biocompatible with C2C12 myoblast cells. According to the literature, this PANI:CSA:PGS composite has not been tested with neural cells to date.

PANI can be blended with PCL to produce conductive fibers suitable for biomedical applications. Previous studies have already shown the potential application of conductive fibers made of PCL:PANI blend as gas sensors, where high conductivities were obtained (8 × 10^−2^ S cm^−1^) [122]. Our group was the first to systematically study the effect of a PANI:CSA concentration in PCL using TFE as the solvent for dispersion [123]. Our physico-chemical analysis of the PCL:PANI systems showed a phase segregation between PCL and the PANI:CSA system, suggesting that the electroconductivity of PANI:CSA would not be significantly affected by PCL. We also showed that the obtained PCL:PANI fibers are soft and more fragile due to PANI:CSA disruption of the PCL packaging. We observed a steady electroconductivity increase until the PANI content totaled 5% (sample composition 95:5, σ = 4.28 × 10^−2^ S cm^−1^), after which a plateau was reached until the PANI content was raised to 12% of the fiber content (sample 88:12, σ = 7.7 × 10^−2^ S cm^−1^). The plateau was attributed to the presence of segregated microscopic PANI:CSA aggregates that disrupted the charge transfer. In a second work [37], we optimized the solvent system used for dispersing PCL:PANI by mixing equal parts of TFE and HFP (TH55). Therefore, the electroconductivity of samples produced in a 50% humidity atmosphere increased by 100× from 8.4 × 10^−4^ S cm^−1^ (sample TFE) to 1.9 × 10^−1^ S cm^−1^ (sample TH55), in a pair with 1.9 × 10^−1^ S cm^−1^ for the sample HFP. Moreover, the obtained PCL:PANI fibers had a larger diameter (373 nm for TH55 vs. 190 nm for TFE), were softer (ε = 1.6 MPa for TH55 vs. ε = 4.8 MPa for TFE and ε ≈ 10 MPa for PCL fibers), and were more hydrophilic (θ = 45° for TH55 vs. θ = 87° for TFE). We attribute this modification of the properties to the pseudo-doping properties of HFP, and we hypothesized that the PANI chains (in PANI:CSA) adopted an expanded coil conformation, which favored chain–chain contact and facilitated electron transfer through hopping. This strategy was employed by our group in the design of biodegradable and electroconductive coaxial fibers [38]. The coaxial structure was obtained using PGS as the core polymer and the PC:-PANI blend (TH55) as the shell/cladding. The obtained coaxial structure prolonged the stability of the electroconductivity of the fibers for up to 21 days in PBS. PANI:CSA was found to retard fiber degradation by lipase, potentially due to enzyme adsorption by electrostatic interaction [124,125]. Other groups have since developed further work on PCL:PANI blends for tissue-engineering applications, including the development of topographic structures [126] and, in the work of Licciardello and colleagues [127] highly hydrophilic and biocompatible fibers after plasma treatment.

The use of PCL:PANI blend for neural cell cultures was first reported by Yang and colleagues using PC12 cells. They tested two concentrations (1% and 3%) and observed constant proliferation for seven days, with normal cell morphology for both samples. Moreover, PC12 neural differentiation was enhanced in PCL:PANI fibers. Our group was the first to test the performance of PCL:PANI fibers with the human-derived neural stem cell lineage ReNCell-VM. Increasing PANI concentration on the fibers did not affect the scaffolds’ biocompatibility, since NSCs were able to attach and interact with the substrate, proliferate, and keep their spindle/stretched morphology. What is more, NSCs on PCL:PANI samples presented higher growth rates than on pristine PCL fibers and acquired a stellar/spindle morphology [123] In a following work, we also demonstrated that PCL:PANI fibers are suitable for the neural differentiation of both ReNCell-VM [37] (monoaxial fibers) and neural progenitors derived from patient-derived iPSCs (coaxial fibers) [38], attesting their suitability for clinical applications.

Other studies addressed the use of PCL:PANI fibers for other relevant tissue-engineering applications [128,129]. One of the first studies addressing the biocompatibility of the PCL:PANI blend, using the C2C12 myoblast cell line, was published by Ku and colleagues [128]. PCL and PANI were blended before electrospinning. Although electroconductivity values were not reported, cyclic-voltammetry measurements showed the redox potential of the fibers and an increasing intensity response with PANI concentrations. Both random and aligned PCL:PANI fibers supported proliferation and stimulated the differentiation of C2C12 cells, and the PANI concentration was correlated with the increasing number of myotubes formed and the higher expression of final differentiation genes such as Tropoin T and MHC. In another work, Li and colleagues [129] synthetized PANI directly on PCL 7% fibers, and a high conductivity value for the fibers was obtained (6.71 × 10^−3^ S cm^−1^). The initial biocompatibility study with human umbilical-vein endothelial cells (HUVECs) showed successful adhesion of the cells to the substrate after 24 h of culture, showing that the material had good biocompatibility. Further studies with electrical stimulation for five days showed an increase in the proliferation performance of the cells on PCL:PANI fibers when compared with pristine fibers, with an increase in the voltage values applied (200, 300 and 400 mV cm^−1^). Interestingly, HUVECs were better able to spread on PCL:PANI fibers than on pristine PCL fibers, and this effect was enhanced by electrical stimulation. Finally, Wibowo and colleagues [65] reported the use of a PCL:PANI blend, with up to 2% of PANI content, for the 3D bioprinting of scaffolds for potential bone-tissue-engineering applications.

PANI can also be used to create blends for hydrogels. PANI is not water soluble and therefore its incorporation in hydrogels depends on the direct coating or dispersion of PANI nanoparticles into the hydrogel, impairing current distribution through the material. Nevertheless, works in the literature describe the development of PANI-based hydrogels with favorable mechanical properties for neural tissue engineering. Stejskal and colleagues [130] developed a PANI cryogel where poly(aniline) was polymerized in situ inside a frozen (−20 °C) PVA (poly(vinyl alcohol) or poly(vinylacetate) solution. The result is a highly porous PANI:PVA cryogel, which is hydrophilic, soft (9.7 kPa) and electroconductive (≈10^−3^ S cm^−1^). Humpolicek and colleagues [68] successfully tested the biocompatibility of PANI:PVA cryogels using ESCs, ES-derived cardiomyocytes, and mice neural progenitors. Neural progenitor cell adhesion and spreading onto the material was possible, despite the low adhesion values obtained. This new materials presentation will surely enable a more homogeneous current distribution, suitable for the electrical stimulation of neural cells.

### 3.4. Electrical Stimulation of Neural Cells on PANI:CSA-Based Platforms

The conductivity and biocompatibility of PANI justify its use in the electrical stimulation of neural cells. The first electrical stimulation studies using PANI-based scaffolds were performed by Ghasemi-Mobarakeh and colleagues [47]. The authors designed electrospun PCL:gelatin:PANI-blend fibers produced for use as nerve guides. The obtained fibers were electroconductive (conductance of 2 × 10^−8^ S), biodegradable, and biocompatible. Finally, NSCs were cultured under electrical stimulation (DC, 1 V cm^−1^) for 60 min and their growth profile and morphology post-stimulation (at one, three and five days) were evaluated. Cell numbers through five days of cell culture post-stimulation were consistently higher and neurite length was higher (30 µm vs. 22 µm) than the non-stimulated samples. According to the authors, electrical stimulation caused depolarization of the cell membrane and induced the effects. Additionally, PANI incorporation did not induce visible effects on NSC proliferation or compromise cell viability after electrical stimulation.

Wang and colleagues [104] reported the use of a PAN:PANI:nickel nanoparticles for the development of an electroconductive hydrogel for the electrical stimulation of Schwann cells. The authors observed an increase in cell proliferation when an electrical field of 100 mV cm^−1^ was applied for one hour per day for five days. A similar influence of the low-intensity electrical fields on NSC proliferation on PANI-based hydrogels was observed by Xu and colleagues [48]. From the three values of voltage tested (15, 35 and 75 mV), NSC proliferation peaked when 15 mV were applied for six hours a day and for seven days. Nevertheless, all three stimulation conditions improved NSC proliferation when compared to the non-stimulated hydrogel.

Zhang and colleagues [70] show the potential of using a dual stimulation system to enhance neural cell differentiation of PC12 cells using PANI-based scaffolds. The authors developed electroconductive poly(L-lactic) acid (PLLA).SF.PANI-blend monoaxial fibers and succeeded in increasing both the length (0.04 vs. 13.9 µm) and number of neurite-positive cells (1.5 vs. 29.7%) by using an electrical field of 100 mV cm^−1^ (one hour per day, five days). Cell alignment with the electrical field was also observed. The blend was used to prepare coaxial fibers for the encapsulation and controlled release of NGF under electrical stimulation. PC12 neurite extension and numbers per cell were further enhanced when the co-axial fibers were used concomitant with electrical stimulation and NGF release.

Our group has also developed PANI-based electrospun fibers for neural-differentiation studies under electrical stimulation. In a first approach [37], ReNCell-VM (hNSCs) were cultured and left to proliferate for four days on randomly oriented PCL:PANI monoaxial fibers, followed by another four days of differentiation. In both stages, electrical stimulation of 1 V cm^−1^, 100 Hz, for 12 h a day (AC stimulation) was applied. The high-intensity electrical field induced cell alignment at the end of both the proliferation and differentiation stages. Moreover, it was observed that electrical stimulation induced a statistically significant upregulation of the neural markers *Dcx*, *Map2* and S100 calcium-binding protein B (*S100β*). In a second study [38], electroconductive and biodegradable PGS/PCL:PANI fibers were used for the electrical stimulation of hiNPCs, obtained after the neural induction of patient-derived iPSCs. The electrical stimulation protocol used was 1 V cm^−1^, 100 Hz for 12 h a day, but instead used pDC stimulation to maximize neuronal differentiation [30]. Electrical stimulation was found to induce an increase in the axon diameter and alignment, as well as increased expression of specific markers associated with (1) cytoskeleton remodeling; (2) general/functional maturation; and (3) a switch in the neural population present from dominant-inhibitory (GABAergic +) to dominant-stimulatory (Glutamatergic +). These results open doors to new strategies for treating neurological diseases.

PANI-based scaffolds have also been tested for in vivo regeneration of peripheral nerves. Wu and colleagues [131] developed a sensor device made of hydroxyethyl cellulose:soy protein isolate:PANI sponge, suitable for man–machine interfaces due to the high electrical signal sensitivity and low inflammatory response after implantation. Electroconductivity of the best scaffolds gradually decreased for 40 days, but these presented favorable mechanical behavior, even after 14 days in PBS, that were deemed suitable for use as a pressure sensor. This sponge was later processed in a nerve conduit used for in vivo sciatic nerve regeneration under electrical stimulation (60 min every 2 days, 14 days) [132]. Rats treated with a combination of the scaffold and electrical stimulation scored higher, similar to the autograph on the sciatic functional index, indicating functional recovery. Three months after surgery, the regenerated nerve was found to be electrophysiologically functional, presented a higher degree of myelinization and a larger axon diameter. Overall, this study demonstrates the potential clinical applications of PANI-based scaffolds.

PANI can also be used in the design of nanoparticles suitable for performing wireless electrical stimulation for cells that internalize them. Kim and colleagues [133] developed gold-doped PANI nanoparticles suitable for internalization by human mesenchymal/stromal stem cells (hMSCs). Nanoparticle uptake allowed for higher calcium internalization by the cells and maximized the response to electrical stimulation (one pulse, 500 V, three days). hMSCs subjected to the electrical stimulation protocol evidenced a change in phenotype to a neuron-like morphology. The phenomenon was accompanied by (1) increased neurite length; (2) higher expression of neural markers (e.g., TUJ1, MAP2); and (3) upregulation of genes associated with neurogenesis, cell proliferation, apoptosis, DNA repair and neural differentiation. Despite the short duration of the assay and the absence of functionality tests of the cells, this work indicated a positive synergy between the electroconductive materials and wireless electrical stimulation for neural applications.

### 3.5. Improving the Bioactivity of the PANI:CSA System

The functionalization of biomaterials to increase their bioactivity is another strategy to improve their applications in tissue engineering. This opens the way for the material’s surface properties to be tailored to the target tissue, allowing cell recognition, boosting biocompatibility, and allowing for a more optimized cell metabolism. Various strategies can be employed to improve cell responses to the electrical field, including: the (1) immobilization of active molecules on a materials surface; (2) the modulation of substrate topography; and (3) the encapsulation and release of small/large electro-facilitator molecules, such as BDNF or ion-channel modulators.

For the design of neural-friendly biomaterials, functionalization with glycosaminoglycans (GAGs) can potentiate the effect of electrical stimulation on neural cells. Erskine and McCaig [72] studied the effect of GAGs on neurite growth of *Xenopus* sp. nerve cells under electrical stimulation. Cells were cultured for two to four hours for neurites to begin to sprout, and the supplementation of culture media with GAGs enriched in chondroitin-4-sulfate (CS-4S), chondroitin-6-sulfate (CS-6S), dermatan sulfate (DS) or keratan sulfate (KS) was performed during the five-hour stimulation (50–133 mV mm^−1^). Differences were found for neurite growth. Namely, CS-6S (10 µg mL^−1^) promoted the neurites to turn to the cathode, whereas CS-4S and DS did not have any effect and KS reduced neurite turning. When higher electrical fields were used, CS-6S continued to promote neurite turning to the cathode but CS-4S decreased neurite turning instead. Concerning neurite growth rates, no changes were observed with GAG supplementation. Interestingly, disaccharide units of CS-4S, CS-6S, DS or KS did not produce any effect on neurite turning. This work suggests that GAGs have a positive effect on neurite directionality, possibly through a direct interaction with the growth cone of the cells. As such, chondroitin-sulfate moieties, or those from other GAGs, could be used to modulate neural cell morphology and growth [134]. Recently, Thompson and colleagues demonstrated the different effects of extracellular-matrix (ECM) molecules isolated from differentiated protoplastamic (grey matter) and fibrous (white matter) astrocytes on axonal growth and guidance [135]. This strategy can be used together with electrical stimulation to improve differentiation or to guide neural cell neurites to specific target regions.

Topography influences neural cell differentiation by influencing cytoskeleton organization and by concentrating growth/differentiation factors near the differentiating cells. Topographic effects can be achieved when, for example, ridges and grooves are produced (by lithography, two-photon printing) in the flat film of a biomaterial. In their work, Tan and colleagues [136] focused on testing the influence of three kinds of gratings on a dopaminergic differentiation protocol for murine NPCs, these included biomaterials with 2 µm, 250 nm and 2 × 2 × 2 µm with interior 250 × 250 × 250 nm (hierarchical grafting). In initial studies, the immunofluorescence of TUJ1 (early neural marker), MAP2 (intermediate neural marker) and glial fibrillary acidic protein (GFAP) (early astrocyte marker) were analyzed. All three gratings showed higher expressions of TUJ1 protein when compared to GFAP, suggesting that more cells were differentiating into neurons, and MAP2 expression was higher in the 2 µm gratings. Elongation and cell alignment was also higher for the 2 µm and hierarchical gratings. When tyrosine hydroxylase (TH), an enzyme involved in dopamine synthesis, was targeted (dopaminergic neurons), high expression values were also found in the same structures, which were not significantly different even after normalization with TUJ1 values. This suggests that the three major tested structures enhanced dopaminergic neuron differentiation (at the protein level) compared to an unpatterned control. However, gene expression analysis revealed significant differences: tyrosine hydroxylase cDNA (*Th*) expression was higher in the hierarchical samples and the expression of pituitary homebox 3 (PITX3), a transcription factor important for dopaminergic neuron survival during development, was higher in the 2 µm gratings. Overall, the work shows the positive effect of topography on neural cell differentiation, especially for constructs with higher spacings (2 µm and hierarchical). Not only do these patterned materials provide more surface area for cells to adhere to, but they also initiate mechano-sensing mechanisms that can potentially lead to changes in the expression of the genes involved in cell differentiation.

Yang and colleagues [137] investigated the additive effect of nanopatterning and electroconductive substrates on myogenic and neural cell differentiation. In one study, gold (2.0 × 10^5^ S m^−1^) or titanium (1.2 × 10^3^ S m^−1^) were electro-evaporated on nanopatterned (800 nm width × 600 nm height) poly(urethane acrylate) (PUA), dimensioned to match the naturally aligned collagen fibers in the muscle tissue. The results they obtained indicate that C2C12 cells align with these designed nanopatterns and myotube length and width, and that the fusion index of myocytes and the expression of differentiation genes increase with the increasing substrate electroconductivity. Both these effects were found to act synergically in promoting cell differentiation in myocytes.

In another study by Yang and colleagues [42] the separate and combined effects of nanopatterning, the electroconductive layer (titanium), and electrical stimulation in neural stem cell differentiation were examined. A positive effect of electroconductivity on cell differentiation was observed. This was attributed to the ability of the titanium layer to short-circuit natural electrical signals across it (passive stimulation), leading to increased protein adsorption, thus affording a more cell-friendly scaffold. Different groove–ridge distances were tested (150, 200, 250 and 300 nm) and only 150 and 200 nm distances were found to increase the overall neural differentiation phenotype of the cells (increasing TUJ1 expression, increasing body length and increasing neurite formation). The process was mediated by mechano-transduction and was associated with:More contact points for the local adhesion of the cells inducing integrin-binding and clustering, with the consequential increase in integrin-binding domains causing focal adhesion kinase (FAK) activation;Actin rearrangement and the modulation of cell contractibility by actomyosin, mediated by the rho-associated protein kinase (ROCK) pathway. The expression of MAP2 levels in hESCs was also important;Consequential activation of the mitogen-activated protein kinase (MEK)-ERK pathway, and the enhancement of neuronal differentiation.

They concluded that cytoskeletal organization and actomyosin contractibility were paramount and enhanced hNSC differentiation. This can be promoted when cells are cultured on nanopatterned substrates, and further enhanced when the substrate is electroconductive. Similar results concerning patterned scaffolds and their influence on cell differentiation were found in other studies [138,139,140].

Patterning results from fiber diameter and alignment using electrospun fibers of poly(ethersulfone) (PES) by Christopherson and colleagues [141] showed the effect of fiber diameter on rat neural stem cell differentiation. The obtained PES fibers had average diameters of 273 ± 45, 749 ± 153 and 1452 ± 312 nm. One main change observed in cell growth was a higher proliferation on the tissue culture plate and fibers with smaller diameters. Fiber diameter did not affect the amount of adsorbed laminin, and nestin expression was observed to be equal for all samples. Therefore, the most plausible factor responsible for these cell growth properties was topography. Moreover, when the differentiation profile was analyzed, the cells were found to express higher levels of TUJ1 (neurons) and lower levels of nestin (neural stem cells) and GFAP (astrocytes) on fibers with a larger diameter (749 nm) than on thinner ones. Wang and colleagues [73] established a relation between fiber diameter and neurite growth and migration of chick dorsal root ganglia-derived Schwann cells. For the three types of PLLA fibers tested (293 ± 65, 759 ± 169 and 1325 ± 383 nm), a higher diameter was associated with a higher neurite length and a greater migration distance of Schwann cells.

Similar results for cell alignment were obtained by Johnson and colleagues [142] for aligned PLLA fibers when rat astrocytes and neurons were co-cultured. Interestingly, it was the topography induced by the presence of electrospun fibers and not the fiber diameters alone that was a critical factor in increasing the neuroprotective properties of astrocytes towards co-cultured neurons. This phenomenon was mediated by the increased expression of the glutamate receptor GLUT-1 in astrocytes, responsible for glutamate removal from the extracellular media. Overall, aligned fibers enhance cell elongation and promote faster regeneration of peripheral nerve cells. This was also shown by Chang and colleagues [143] who found a spiral scaffold to be more effective for nerve repair/regeneration of rats when aligned fibers were used as guiding cues.

## 4. Challenges, Opportunities and Conclusions

In the final section of this review, we discuss how PANI:CSA platforms can be improved and provide possible future development avenues. The first is by increasing the chemically stability of the doping. PANI:CSA platforms that are suitable for ES must also be stable in aqueous media at physiological conditions (e.g., 37 °C, 5% CO_2_, pH = 7.4; ionic strength of 90 mM), without, and/or in the presence of relevant enzymes. In most of the studies reviewed here, such screening is mostly overlooked in favor of the maximization of electroconductivity. It is arguable that the mechanism of transduction of ES into a biological response is solely dependent on the electron or ionic conductivity of an electroconductive polymers [144,145], thus potentially making this a non-issue. However, the natural de-doping of PANI:CSA that occurs in aqueous solutions [38,106] limits the reliability and applicability of PANI:CSA-based platforms for the long-term ES of neural cells (e.g., 28 days). In our past work, we addressed this issue by developing coaxial PGS/PCL:PANI fibers with prolonged stability of electroconductivity in a saline solution and in the presence of the enzyme lipase [38]. We attribute this improved stability to PGS, which has previously been shown to improve the stability of PANI:CSA, as studied by Qazi and colleagues [75]. However, we believe this is not enough. We believe that the PANI:CSA system should be chemically modified to improve its stability in aqueous media, as its instability results from CSA dissolution in aqueous media, especially at pH values close to 7. We propose that CSA and/or other relevant doping agents (e.g., hyaluronic acid) should be chemically linked to a support material (e.g., PCL, PLLA). Since PANI is not water soluble, such a strategy would allow for the immobilization of PANI and improve the stability of PANI:CSA, while increasing its versatility for additive manufacturing.

Some groups have reported the development of water-soluble and self-doped PANI, which might be advantageous for processing in adequate ES platforms [146,147]. However, the production of PANI derivates involves complex synthesis and purification steps that are not easily reproducible. As such, another, easier, strategy to improve the stability of PANI:CSA is to change the doping agent used. Other groups developed composites with PANI doped with water-soluble molecules/polymers (e.g., hyaluronic acid, PLLA) with the ability to dope PANI through free carboxylic groups (pKa = 4–5) [148,149]. CSA has a pKa ≈ 1.2 [150], and potential candidates for replacement should have a similar or lower pKa value, while also allowing for an easy processability into scaffolds. An interesting candidate class for replacing CSA would be polyelectrolytes. Among these, PSS (pKa = 1.0) [151] is the most interesting due to its wide reported use with PEDOT, another biocompatible electroconductive polymer. In the PEDOT:PSS system, PSS is also responsible for improving the water dispersibility of PEDOT. It is possible that a similar PANI:PSS system would improve the water dispersibility of PANI and greatly reduce the use of organic solvents. Similar to PEDOT:PSS, cross-linking with (3-glycidyloxypropyl)trimethoxysilane (GOPS) [152] and divinyl sulfone (DVS) [153], or even sulfuric acid treatment [154], could also be employed to improve water resistance and improve the electrical properties of PANI. Other polyelectrolytes are also possible, including biological-derived ones such as polyglutamic acid (pKa = 4.86) [155], hyaluronic acid (pKa = 3.0) [149], or even desoxyribonucleic acid (DNA) (pKa ≈ 2.0) [156].

Finally, if de-doping of PANI:CSA cannot be avoided, it can be harnessed for other applications. In the work of Bhattacharya and colleagues [157], PANI was doped with the antibiotic chloroxylenol and used for the production of poly(ethylene oxide) (PEO):PANI:chloraroxylenol electrospun fibers (3.2 S cm^−1^). The released chloroxylenol greatly improved the antibacterial properties of PANI:PEO fibers against Gram-positive and Gram-negative bacteria. Other, more sophisticated, systems could be developed for the controlled release of immobilized drugs through the application of specific voltage values. This is similar to work described by Sun and colleagues [44].

In conclusion, the manipulation of bioelectricity through electrical stimulation has dramatic effects on the phenotype of neural cells. When physiologically relevant electrical fields are applied, several biomolecular cascades are activated, resulting in an improvement of NSC differentiation. Electroconductive materials, such as PANI:CSA, can be used in the development of bioelectroactive composites and scaffolds suitable for directing electrical stimulation to the cultured cells. Pseudo-doping agents and smart composite design can be employed to maximize electroconductivity and stability. The biocompatibility of PANI:CSA ensures its potential clinical use in long-term applications, including transplantation. The electrical stimulation of neural cells on PANI-based scaffolds improves differentiation as well as when other electroactive materials are used, including PEDOT:PSS and PPY. Improving bioactivity through the immobilization of biomolecules and/or processing PANI-based constructs into topographic/3D scaffolds is possible and desirable for a more efficient neural cell differentiation. Overall, PANI:CSA continues to be a relevant system for the design of platforms/scaffolds for the therapy of neurological diseases using electrical stimulation and tissue engineering. Further studies on PANI:CSA should focus on improving doping stability, potentially through cross-linking and/or a change of CSA for a bulkier doping agent. This would also increase the diversity of scaffolds that could be obtained.

## Figures and Tables

**Figure 1 polymers-15-02674-f001:**
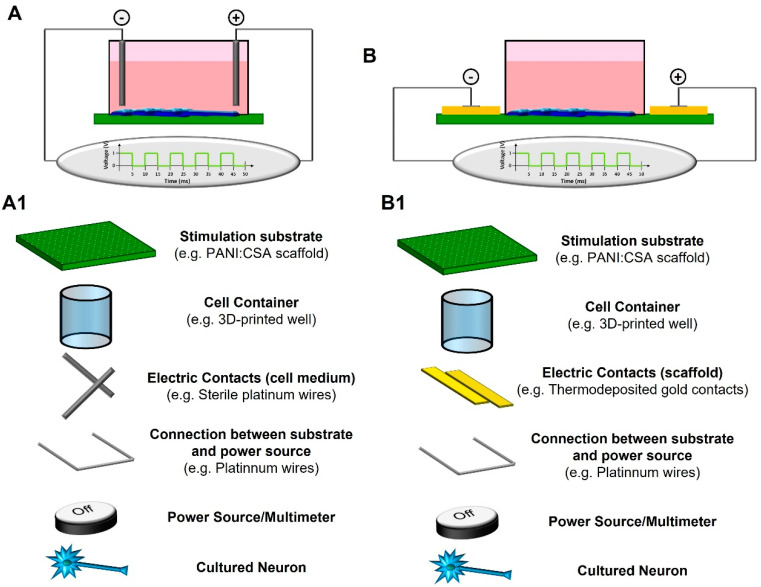
Schematics of the basic bioreactor for the electrical stimulation of neural cells: (**A**) culture medium bioreactor, and (**A1**) respective composition; and (**B**) supporting material bioreactor, and (**B1**) respective composition.

**Figure 2 polymers-15-02674-f002:**
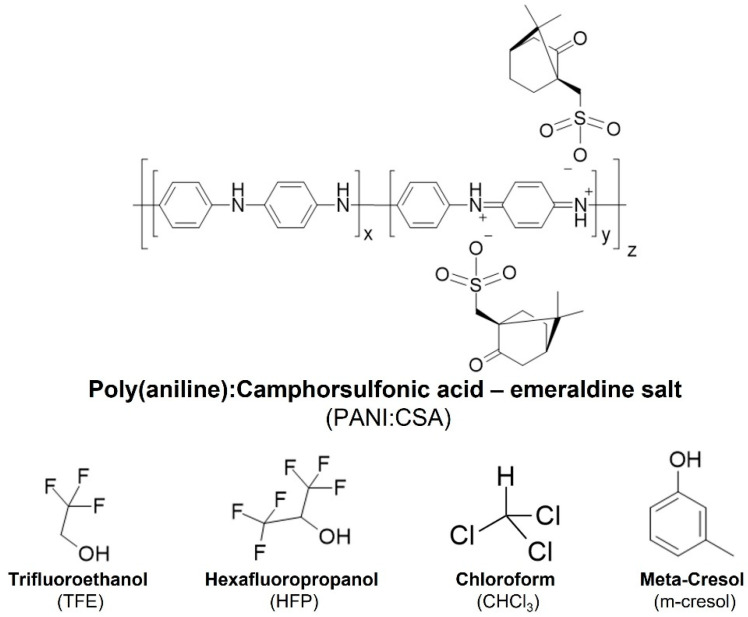
Chemical structure of poly(aniline) doped with camphorsulfonic acid (PANI:CSA) and solvents suitable for its processability.

## Data Availability

Not applicable.

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
