# Peer review of "Designing Electrical Stimulation Platforms for Neural Cell Cultivation Using Poly(aniline): Camphorsulfonic Acid"

_polymers, 2023, doi:10.3390/polym15122674_

Round 1

Reviewer 1 Report

The review article by Garrudo and coworkers deals with the electrical stimulation of neural cells through the electroconductive polymer Poly(aniline):camphorsulfonic acid (PANI:CSA).

The review article seems to be very long, although the Authors define it a "concise and focused review" (line 118). It is strongly suggested to shorten the text, which now resembles an exaustive book chapter, rather than a concise review. In this Reviewer opinion, several parts of sections 2.1, 2.2 and 2.3 could be shortened, because they contain well-known concepts related to bioelectrical signaling in cells  (i.e. ion channels, voltage-dependent channels) or out-of-focus considerations (i.e. epithelial cell bioelectricity, eye cells bioelectricity).

Lines 576-588 describe the figure 1, as the figure caption does. Therefore, it is possible to remove this numbered list and shorten this paragraph, without losing important information.

Sometimes, too much details about the cited references are reported. Conversely, synthetic information should be provided to the reader.

The final considerations should be a critical overview of the future development of platforms for electrical stimulation of neural cells. In this scenario, the impact of PANI-based technologies and their improvement should be compared to other polymers cited in the previous sections. It is suggested to show in much more detail "what is actually next" in this research field.

As a minor issue, please delete redundant references, inserting only those strictly needed.

A revision of the whole review is suggested, checking English language and avoiding typos (i.e. line 323, 333). Define all the acronyms within the manuscript at first appearance (i.e. HBEGF, line 635).

Author Response

The review article by Garrudo and coworkers deals with the electrical stimulation of neural cells through the electroconductive polymer Poly(aniline):camphorsulfonic acid (PANI:CSA).

Thank you for your critical analysis on our manuscript, we appreciate the detailed revision of this manuscript.  We believe we have addressed the issues raised and implement most of the recommended changes and improved the manuscripts quality.  We will answer each one point-by-point.

The review article seems to be very long, although the Authors define it a "concise and focused review" (line 118). It is strongly suggested to shorten the text, which now resembles an exaustive book chapter, rather than a concise review. In this Reviewer opinion, several parts of sections 2.1, 2.2 and 2.3 could be shortened, because they contain well-known concepts related to bioelectrical signaling in cells  (i.e. ion channels, voltage-dependent channels) or out-of-focus considerations (i.e. epithelial cell bioelectricity, eye cells bioelectricity).

We thank the reviewer for pointing this issue out, and we understand this might be misleading.  We have replaced the previous definition (old version line 118) concise and focused for (new version, line 104) broad and multidisciplinary.  We believe that it is important to understand the biology/bioelectricity and design fundaments of electrical stimulation (ES) platforms to fully understand the applicability of the poly(aniline):camphorsulfonic acid polymeric system in their design.  Moreover, the off-topics mentioned by the reviewer (e.g. epithelial cell bioelectricity, eye cells bioelectricity) were purposely inserted into the text to complement the information provided.  ES studies with neural cells (e.g. neurons, astrocytes, glial cells) are limited, and as such other close biological models are necessary to understand the bioelectricity principles that govern cell patterning and selective differentiation.  This is especially relevant in the eye and its respective epithelial cells, which are also an integral part of the central nervous system [1–3] .

We have previously disclaimed this fact in lines 107-109, where the reviewer and other readers can read (old version, lines 121-125):

Although neural cells are the focus of this review, including neural cell bioelectricity and basic neural cell biology/biochemistry, we also consider other off-topic case-studies, when deemed necessary, to better understand the manipulation of bioelectricity and design of electrical stimulation platforms (e.g. eye/retinal epithelial cell bioelectricity).

We hope the reviewer understands our option of not shorten the manuscript, as kindly suggested.  We believe the manuscript, as it is, will provide the readers with a reading option of higher value.  The readers will still find most of the relevant studies of PANI:CSA summarized and extensively discussed and integrated with other relevant studies.  At the same time, basic biological/bioelectricity concepts are discussed for the readers to best understand the implications of ES on neural cells and the optimal physico-chemical requirements of PANI:CSA platforms for that to occur.

Lines 576-588 describe the figure 1, as the figure caption does. Therefore, it is possible to remove this numbered list and shorten this paragraph, without losing important information.

We thank the reviewer for the kind suggestion and understand that some of the information contained in Figure 1 might be repeated in the text (old version, lines 576-588).  We agree with the reviewer and have decided to remove the aforementioned information from the main text.

Sometimes, too much details about the cited references are reported. Conversely, synthetic information should be provided to the reader.

We thank the reviewer for the observation and agree that the manuscript provides very detailed information about the studies.  We have thoroughly revised the manuscript and decided to keep the level of detail as it is.  Our goal with this manuscript is to provide the readers with a complete analysis of the literature, enough to educate and give the tools necessary to develop their own ES platforms based on PANI:CSA.  We believe we achieved this with the manuscript in its current form.  We also agree with the reviewer that this makes our review manuscript resemble more of a “Book Chapter”, which we see as a positive critique on the quality of our work.

The final considerations should be a critical overview of the future development of platforms for electrical stimulation of neural cells. In this scenario, the impact of PANI-based technologies and their improvement should be compared to other polymers cited in the previous sections. It is suggested to show in much more detail "what is actually next" in this research field.

We thank the reviewer for the great suggestion, and we agree that a proper discussion of the topic should be expanded and summarized in a separate section of the manuscript.  We took this opportunity to modify section 4 to create such a section.  You can now read (new version, line 880-919):

  1. CHALLENGES, OPPORTUNITIES AND CONCLUSIONS.

In the final section of this review, we discuss how PANI:CSA platforms can be improved and provide possible future development alleys.  The first one is increasing the chemically stability of the doping.  PANI:CSA platforms that are suitable for ES must also be stable in aqueous media at physiological conditions (e.g. 37 oC C, 5% CO2, pH = 7.4; ionic strength of 90 mM), without and/or in the presence of relevant enzymes.  In most of the studies reviewed here, such screening is mostly overlooked in favor of maximization of electroconductivity.  It is arguable whether the mechanism of transduction of ES into a biological response is solely dependent on the electron or ionic conduc-tivity of an electroconductive polymers  [146,147] , thus potentially making this a non-issue.  However, the natu-ral de-doping of PANI:CSA that occurs in aqueous solutions  [38,107]  limits the reliability and applicability of PANI:CSA-based platforms for long-term ES of neural cells (e.g. 28 days).  In our past work we addressed this issue by developing coaxial PGSPCL:PANI fibers with prolonged stability of electroconductivity in saline solution and in the presence of the enzyme lipase  [38] .  We attribute this improved stability to PGS, which has  previously shown to improve the stability of PANI:CSA, as studied by Qazi and colleagues  [76] .  However, we believe this is not enough.  We believe PANI:CSA system should be chemically modified to improve its stability in aqueous me-dia, as its instability results from CSA dissolution in aqueous media, especially at pH values close to 7.  We propose CSA and/other relevant doping agents (e.g. hyaluronic acid) to be chemically linked to a support material (e.g. PCL, PLLA).  Since PANI is not water soluble, such strategy would allow to immobilize PANI and improve the stability of PANI:CSA, while increasing its versatility for additive manufacturing.

Some groups have reported the development of water-soluble and self-doped PANI, which might be advanta-geous for processing in adequate ES platforms [148,149] .  However, the production PANI-derivates involves com-plex synthesis and purification steps and are not easily reproducible.  As such, another easier strategy to improve the stability of PANI:CSA is to change the doping agent used.  Other groups developed composites with PANI doped with water-soluble molecules/polymers (e.g. hyaluronic acid, PLLA) with the ability to dope PANI through free carboxylic groups (pKa = 4-5) [150,151] .  CSA has a pKa ≈ 1.2 [152] , and potential candidates for replacement should have a similar or lower pKa value, while also allow an easy processability into scaffolds.  An interesting candidate class for replacing CSA would be polyelectrolytes.  Of these, PSS (pKa = 1.0) [153]  is the most interest-ing one due to its wide reported use with PEDOT, another biocompatible electroconductive polymer.  In the PE-DOT:PSS system, PSS is also responsible for improving the water dispersibility of PEDOT.  It is possible that a sim-ilar PANI:PSS system would improve the water dispersibility of PANI and greatly reduce the use of organic sol-vents.  Similarly to PEDOT:PSS, cross-linking with (3-glycidyloxypropyl)trimethoxysilane (GOPS) [154]  and divinyl sulfone (DVS) [155] , or even sulfuric acid treatment [156] , could also be employed to improve water re-sistance and improve the electrical properties of PANI.  Other polyelectrolytes are also possible, including biologi-cal-derived ones including polyglutamic acid (pKa = 4.86) [157] , hyaluronic acid (pKa = 3.0) [151] , or even desoxy-ribonucleic acid (DNA) (pKa ≈ 2.0) [158] .

Finally, if de-doping of PANI:CSA cannot be avoided, it can be harnessed for other applications.  In the work of Bhattacharya and colleagues [158] , PANI is doped with the antibiotic chloroxylenol and used for the production of poly(ethylene oxide) (PEO):PANI:chloraroxylenol electrospun fibers (3.2 S cm-1).  The released chloroxylenol greatly improved the antibacterial properties of PANI:PEO fibers against Gram-positive and Gram-negative bacte-ria.  Other more sophisticated systems could be developed for the controlled release of immobilized drugs through the application of specific voltage values.  This is similar to what was described by Sun and colleagues [44] .

As a minor issue, please delete redundant references, inserting only those strictly needed.

We thank the reviewer for this suggestion and we agree that only strictly necessary references should be used in the manuscript.  We have thoroughly revised the manuscript and found all references contained are necessary and irreplaceable.

We want to highlight that the same work of literature can be references more than once.  This is because these works contain multidisciplinary results which must be discussed in appropriate and separate sections.  For example, the work of Song and colleagues [4]  is discussed twice, once in section 2.3 (new version, lines 282-292), where we discuss the consequences of ES on neural stem cell (NSC) secretome, and again in section 2.4. (new version, lines 465-475) where we discuss the influence of 2D/3D scaffolds for effective ES of neural cells.  At both times, a small recapitulation of the work is given to the reader for better understanding of the discussion undertaken.

A revision of the whole review is suggested, checking English language and avoiding typos (i.e. line 323, 333). Define all the acronyms within the manuscript at first appearance (i.e. HBEGF, line 635).

We thank the reviewer for the great suggestion of making a general revision of the overall manuscript.  We agree this greatly improves the quality of the manuscript.  We have conducted a thorough English language checking with the help of native speakers and corrected all the typos and errors found.  We took special attention to the typos highlighted by the reviewer, including:

- (old version, line 323) only enhanced the length of the obtained neurites neurites, while…, you can now read (new version, line 247) …only enhanced the length of the obtained neurites, while…

We have also re-checked all the acronyms used and described their meaning at first appearance.  HBEGF, the acronym for Heparin binding EGF like growth factor highlighted by the reviewer as not being described (old version, line 635; new version, line 466), has been previously defined (old version, lines 376-377; new version, line 285) but not in a correct manner.  We modified the definition for a more correct one.  You can now read (new version, line 285):

…heparin binding epidermal growth factor-like growth factor (HBEGF)…

References:
[1]        M.-C. health perspectives, The eye and visual nervous system: anatomy, physiology and toxicology., Environmental Health Perspectives. (1982).

[2]          AP Polednak, F.-J. Cancer, Brain, other central nervous system, and eye cancer, Cancer. (1995).

[3]        S. Waduthantri, Eye as a window to the brain in central nervous system diseases, Medical J Dr D Y Patil Vidyapeeth. 12 (2019) 376.

[4]        S. Song, D. Amores, C. Chen, K. McConnell, B. Oh, A. Poon, et al., Controlling properties of human neural progenitor cells using 2D and 3D conductive polymer scaffolds, Sci. Rep. 9 (2019) 19565.

Reviewer 2 Report

The authors have collected and evaluated the application of electrical stimulation to neural cells, including fundamentals of bioelectricity and electrical stimulation, how electrical stimulation can be directed to cultured cells using PANI:CSA-based systems, and examples of scaffolds and setups suitable for electrical stimulation. This review can inspire more design ideas of electrical stimulation of cells using electroconductive PANI:CSA platforms/scaffolds for the clinical application. Overall, this is a well-written and well-organized review paper. Therefore, I would like to recommend this review to publish in Polymers. Below are some comments for the authors.

1. The caption of Table 1 should be place on the top of Table 1.

2. The right side of Table 1 was cut off. Please check and revise Table 1.

3. For a review paper, the discussion with Challenges, Opportunities, and Conclusions should be provided at the end of the manuscript. This review would be more impressive, if the authors could provide discussion with Challenges, Opportunities, and Conclusions.

Author Response

The authors have collected and evaluated the application of electrical stimulation to neural cells, including fundamentals of bioelectricity and electrical stimulation, how electrical stimulation can be directed to cultured cells using PANI:CSA-based systems, and examples of scaffolds and setups suitable for electrical stimulation. This review can inspire more design ideas of electrical stimulation of cells using electroconductive PANI:CSA platforms/scaffolds for the clinical application. Overall, this is a well-written and well-organized review paper. Therefore, I would like to recommend this review to publish in Polymers. Below are some comments for the authors.

Thank you for your critical analysis on our manuscript and the positive feedback.  We believe we have addressed the issues raised and implemented most of the recommended changes and improve the manuscripts quality.  We will answer each one point-by-point.

  1. The caption of Table 1 should be place on the top of Table 1.

We thank the reviewer for highlighting this issue and we agree that the caption for Table 1 must be placed at the top of it.  We have addressed this issue and now the caption can be found in the correct place.

  1. The right side of Table 1 was cut off. Please check and revise Table 1.

We thank the reviewer for highlighting this issue, for much valuable information was cut off.  We have addressed this issue, and now Table 1 can be found whole in the text. 

  1. For a review paper, the discussion with Challenges, Opportunities, and Conclusions should be provided at the end of the manuscript. This review would be more impressive, if the authors could provide discussion with Challenges, Opportunities, and Conclusions.
  2. CHALLENGES, OPPORTUNITIES AND CONCLUSIONS.

In the final section of this review, we discuss how PANI:CSA platforms can be improved and provide possible future development alleys.  The first one is increasing the chemically stability of the doping.  PANI:CSA platforms that are suitable for ES must also be stable in aqueous media at physiological conditions (e.g. 37 oC C, 5% CO2, pH = 7.4; ionic strength of 90 mM), without and/or in the presence of relevant enzymes.  In most of the studies reviewed here, such screening is mostly overlooked in favor of maximization of electroconductivity.  It is arguable whether the mechanism of transduction of ES into a biological response is solely dependent on the electron or ionic conduc-tivity of an electroconductive polymers  [146,147] , thus potentially making this a non-issue.  However, the natu-ral de-doping of PANI:CSA that occurs in aqueous solutions  [38,107]  limits the reliability and applicability of PANI:CSA-based platforms for long-term ES of neural cells (e.g. 28 days).  In our past work we addressed this issue by developing coaxial PGSPCL:PANI fibers with prolonged stability of electroconductivity in saline solution and in the presence of the enzyme lipase  [38] .  We attribute this improved stability to PGS, which has  previously shown to improve the stability of PANI:CSA, as studied by Qazi and colleagues  [76] .  However, we believe this is not enough.  We believe PANI:CSA system should be chemically modified to improve its stability in aqueous me-dia, as its instability results from CSA dissolution in aqueous media, especially at pH values close to 7.  We propose CSA and/other relevant doping agents (e.g. hyaluronic acid) to be chemically linked to a support material (e.g. PCL, PLLA).  Since PANI is not water soluble, such strategy would allow to immobilize PANI and improve the stability of PANI:CSA, while increasing its versatility for additive manufacturing.

Some groups have reported the development of water-soluble and self-doped PANI, which might be advanta-geous for processing in adequate ES platforms [148,149] .  However, the production PANI-derivates involves com-plex synthesis and purification steps and are not easily reproducible.  As such, another easier strategy to improve the stability of PANI:CSA is to change the doping agent used.  Other groups developed composites with PANI doped with water-soluble molecules/polymers (e.g. hyaluronic acid, PLLA) with the ability to dope PANI through free carboxylic groups (pKa = 4-5) [150,151] .  CSA has a pKa ≈ 1.2 [152] , and potential candidates for replacement should have a similar or lower pKa value, while also allow an easy processability into scaffolds.  An interesting candidate class for replacing CSA would be polyelectrolytes.  Of these, PSS (pKa = 1.0) [153]  is the most interest-ing one due to its wide reported use with PEDOT, another biocompatible electroconductive polymer.  In the PE-DOT:PSS system, PSS is also responsible for improving the water dispersibility of PEDOT.  It is possible that a sim-ilar PANI:PSS system would improve the water dispersibility of PANI and greatly reduce the use of organic sol-vents.  Similarly to PEDOT:PSS, cross-linking with (3-glycidyloxypropyl)trimethoxysilane (GOPS) [154]  and divinyl sulfone (DVS) [155] , or even sulfuric acid treatment [156] , could also be employed to improve water re-sistance and improve the electrical properties of PANI.  Other polyelectrolytes are also possible, including biologi-cal-derived ones including polyglutamic acid (pKa = 4.86) [157] , hyaluronic acid (pKa = 3.0) [151] , or even desoxy-ribonucleic acid (DNA) (pKa ≈ 2.0) [158] .

Finally, if de-doping of PANI:CSA cannot be avoided, it can be harnessed for other applications.  In the work of Bhattacharya and colleagues [158] , PANI is doped with the antibiotic chloroxylenol and used for the production of poly(ethylene oxide) (PEO):PANI:chloraroxylenol electrospun fibers (3.2 S cm-1).  The released chloroxylenol greatly improved the antibacterial properties of PANI:PEO fibers against Gram-positive and Gram-negative bacte-ria.  Other more sophisticated systems could be developed for the controlled release of immobilized drugs through the application of specific voltage values.  This is similar to what was described by Sun and colleagues [44] .

Reviewer 3 Report

Dear Authors,

The manuscript entitled "Designing electrical stimulation platforms for neural cells cultivation using poly(aniline):camphorsulfonic acid" is a well written review. It contains important information for the readers. 

My only concern is only the format of the table.

The whole manuscript has the quality to continue to the next step of the publication process

Dear Authors,

I do not have any comment to perform for the quality of the English of the manuscript.

Author Response

Comments to the authors:

Dear Authors,

The manuscript entitled "Designing electrical stimulation platforms for neural cells cultivation using poly(aniline):camphorsulfonic acid" is a well written review. It contains important information for the readers. 

Thank you for your critical analysis on our manuscript and the positive feedback.  We believe we have addressed the issues raised and implemented most of the recommended changes and improve the manuscripts quality.  We will answer each one point-by-point.

My only concern is only the format of the table.

We thank the reviewer for highlighting this issue, for much valuable information was cut off.  We have addressed this issue, and now Table 1 can be found whole in the text.

The whole manuscript has the quality to continue to the next step of the publication process

We thank the reviewer for the very positive feedback on our manuscript.

Round 2

Reviewer 1 Report

The manuscript has been significantly improved and is now suitable for publication.